# Recent Advances of Upconversion Nanomaterials in the Biological Field

**DOI:** 10.3390/nano11102474

**Published:** 2021-09-22

**Authors:** Cunjin Gao, Pengrui Zheng, Quanxiao Liu, Shuang Han, Dongli Li, Shiyong Luo, Hunter Temple, Christina Xing, Jigang Wang, Yanling Wei, Tao Jiang, Wei Chen

**Affiliations:** 1Beijing Key Laboratory of Printing and Packaging Materials and Technology, Beijing Institute of Graphic Communication, Beijing 102600, China; gcj2019015220@163.com (C.G.); zhengpengrui9999@126.com (P.Z.); drllqx@163.com (Q.L.); 13552797385@163.com (S.H.); lidongli@bigc.edu.cn (D.L.); luoshiyong@bigc.edu.cn (S.L.); 2Department of Physics, The University of Texas at Arlington, Arlington, TX 76019-0059, USA; hunter.temple@uta.edu (H.T.); Christina.Xing@uta.edu (C.X.); 3Faculty of Applied Sciences, Jilin Engineering Normal University, Changchun 130052, China; 4CAS Center for Excellence in Nanoscience, Beijing Key Laboratory of Micro-Nano Energy and Sensor, Beijing Institute of Nanoenergy and Nanosystems, Chinese Academy of Sciences, Beijing 101400, China; 5Medical Technology Research Centre, Chelmsford Campus, Anglia Ruskin University, Chelmsford CM1 1SQ, UK

**Keywords:** UCNPs, UCL, RE-doped, bio-imaging, PDT, bio-detection

## Abstract

Rare Earth Upconversion nanoparticles (UCNPs) are a type of material that emits high-energy photons by absorbing two or more low-energy photons caused by the anti-stokes process. It can emit ultraviolet (UV) visible light or near-infrared (NIR) luminescence upon NIR light excitation. Due to its excellent physical and chemical properties, including exceptional optical stability, narrow emission band, enormous Anti-Stokes spectral shift, high light penetration in biological tissues, long luminescent lifetime, and a high signal-to-noise ratio, it shows a prodigious application potential for bio-imaging and photodynamic therapy. This paper will briefly introduce the physical mechanism of upconversion luminescence (UCL) and focus on their research progress and achievements in bio-imaging, bio-detection, and photodynamic therapy.

## 1. Introduction

Luminescent materials are functional materials that absorb incident energy and subsequently emit incident energy in photons. Most luminescent materials, including organic dyes and quantum dots (QDs) [1], follow Stokes Law. They absorb high-energy photons and emit low-energy photons (energy reduction), known as downconversion materials [2,3,4,5,6,7]. On the contrary, UCNPs relate to the conversion of long-wavelength NIR light with lower energy into ultraviolet or visible light with higher energy. This process is also known as the anti-stokes process, which is so-called the UCL phenomenon.

When almost all the trivalent rare earth ions (RE^3+^) are excited externally, the rare earth elements of 15 lanthanides, yttrium, and scandium can move between different energy levels, because they have a unique 4f electron configuration energy level, such as shown in Figure 1. In addition, due to the shielding of the secondary 5s^2^5p^6^ shell, the external environment has little effect on rare earth ions [8,9,10,11]. Under near infrared (NIR) laser irradiation, typical lanthanide-doped upconversion nanoparticles exhibit anti-Stokes shifted visible light and ultraviolet light emission, while the autofluorescence background is minimal, and the light scattering of biological tissues is greatly reduced. The reduction in light scattering caused by this near-infrared excitation will make the penetration depth of biological tissues far greater than the penetration depth under ultraviolet or visible light excitation, which makes UCNP have great potential in biological applications [12,13,14,15,16,17,18,19,20,21], including biological tissues imaging [22,23,24], biological detection [25,26,27,28,29,30], photodynamic therapy, and other biomedical fields [31,32,33,34,35,36].

In this review, we briefly describe the luminescence mechanism and synthetic methods of UCNPs. However, the surface of UCNPs obtained through conventional surface modification methods lacks the active groups that are binding to biomolecules, which limits their application in the biological field to a certain extent. Here, we introduce several surface modification methods and systematically discuss the latest progress of UCNPS in biological imaging and photodynamic therapy.

## 2. Mechanism of Upconversion

In general, UCNPs are composed of inorganic crystal matrix and rare-earth ions, which do not constitute the light-emitting energy level. The main function is to provide the lattice structure of rare-earth ions to ensure they have appropriate luminescent conditions. According to current research, there are five upconversion mechanisms: excited-state absorption (ESA), energy transfer upconversion (ETU), cooperative sensitization upconversion (CSU), cross-relaxation (CR), and photon avalanche (PA) [37,38].

### 2.1. Excited State Absorption (ESA)

Excited state absorption (ESA) occurs in the form of continuous absorption of pump photons by a single ion through the trapezoidal structure of a simple multistage system. As shown in Figure 2a, it is realized by a three-level system that continuously absorbs two photons. This mechanism is due to the equal separation degree of E1 and E2, E2 and E3, and the storage capacity of intermediate E2. When an ion is excited from the ground state to the E2 level, because the lifetime of the E2 level is very long, the other pump photon is likely to absorb another photon before it decays to the ground state, so as to promote the ion from the E2 level to the higher E3 level, resulting in the upconversion emission of the E3 level. In order to meet the above process and achieve efficient ESA, the energy states of lanthanide elements need to be arranged in a trapezoid, and only a few lanthanide ions, such as Er^3+^, Ho^3+^, Tm^3+^ and Nd^3+^ have such energy level structure [39]. In addition, it is found that the output wavelength of commercial diode lasers (about 975 nm and/or 808 nm) can well match the excitation wavelength of these energy level structures.

### 2.2. Energy Transfer Upconversion (ETU)

Unlike ESA, Energy transfer upconversion (ETU) involves two identical or different ions. In this photophysical process, ion 1 transitions from the ground state to E2 by absorbing excitation light; then the absorbed photon energy is transferred to the ground state E1 and excited state E2 of ion 2, so that they are excited again, while ion 1 relaxes back to ground state E1. In this process, the quantum yield of upconversion largely depends on the average distance between ion 1 and ion 2, and is mainly determined by the concentration of doped rare earth ions. The ETU process is the most important photophysical process in upconversion luminescence, because so far, the use of upconversion nanoparticles for therapeutic diagnostics and other applications is through the sensitizer Yb^3+^, which has a strong absorption of the excitation light at 975 nm, thus making activators (Er^3+^, etc.) produce more efficient fluorescence emission [40,41,42,43,44,45].

Moreover, the scattering and absorption of biological tissues at 975 nm are relatively small, and no optical interference occurs. Here, Yb^3+^ has a very good effect as a sensitizer because it has a sufficiently large absorption cross-section in the near-infrared region of about 975 nm. In addition, since Yb^3+^ has only two energy levels, its optimal concentration can be maintained at a high level (20–100% for fluorinated nanoparticles) without causing harmful cross-relaxation. To date, most research has focused on the development of Yb^3+^ sensitized upconversion nanoparticles pumped at about 975 nm. Using lanthanide ions themselves as sensitizers, high-efficiency ETU can also be observed in single lanthanide doping systems, for example, long-wavelength 1490 nm excitation of Er^3+^-doped LiYF_4_ [46]; or doping under 1200 nm excitation NaGdF_4_ nanoparticles doped with Ho^3+^ [47]. The use of other sensitizers can be used to quench and enhance the luminous intensity of certain emission bands. For example, Nd^3+^, Ce^3+^, and Ho^3+^ are used as sensitizers to enhance the blue emission band of Tm^3+^, the red emission band of Ho^3+^, and the near-infrared emission band of Tm^3+^, respectively [48,49,50,51].

### 2.3. Cooperative Sensitization Upconversion (CSU)

Cooperative sensitization upconversion (CSU) in Figure 2c is a photophysical process of the interaction of three rare earth ions (two types). Ion 1 and ion 3 usually belong to the same sensitizer, such as Yb^3+^. After being excited by light, ion 1 and ion 3 transition to an excited state. However, ion 1 or ion 3 alone cannot excite ion 2, because their excited state energy levels are quite different, so ion 1 and ion 3 need to be co-excited to produce a virtual excited state energy level that can be compared with the excited state energy level of ion 2. The ion 2 absorbs the energy of cooperative sensitization and emits a higher energy photon. The photophysical process of CSU is often uncommon, because the energy transfer efficiency is low, and the para-virtual pair energy levels in the transfer process are involved. These energy levels must be described by quantum mechanics in higher disturbances. Nevertheless, limiting the excitation to compensate for the low efficiency provides a possibility to achieve high-resolution imaging, which is not possible with other upconversion mechanisms. At present, the CSU mechanism of Yb^3+^/Tb^3+^ [52], Yb^3+^/Eu^3+^ [53], Yb^3+^/Pr^3+^ [54] ion pairs has been reported.

### 2.4. Cross-Relaxation (CR)

The CR process involves two identical or different ions. In the CR process, ion 1 transfers part of the energy of the E2 energy level to ion 2, causing ion 2 to transfer to a higher excited state, while ion 1 returns to a lower energy level through a non-radiative relaxation process (Figure 2d). Although CR is related to the concentration quenching effect, it can be used to adjust the emission spectra of UCNPs. The related studies currently reported include Y_2_O_3_: Yb^3+^/Er^3+^ [55], GdPO_4_:Sm^3+^ [56], KYF_4_:Tb^3+^, Yb^3+^ [57], etc.

### 2.5. Photon Avalanche (PA)

PA includes ESA process and CR process. Photon avalanche (PA) in Figure 2e is a process of upconversion above a certain excitation power threshold. Once excited, the upconversion fluorescence luminescence intensity will increase by orders of magnitude. In addition, the excited state energy levels of rare earth ions are also required to have a relatively high lifetime. In this process, the CR process mentioned above is also required. Ion 2 transitions from E2 back to E1 and releases energy at the same time. Ion 1 absorbs the energy and transitions from E1 to E2, and then transfers the energy to the E1 energy level of ion 2. At this time, the E1 energy level of ion 2 will absorb photons through the ESA process, which increases the population of the E1 energy level exponentially, leading to the PA process. This photophysical process is often uncommon because of the higher excitation power density required. Er^3+^/Yb^3+^ co-doped NaBi(WO_4_)_2_ phosphor produces strong green upconversion luminescence through the photon avalanche process [58]. Under excitation at 980 nm, the seven-photon PA upconversion (UC) behavior of Er^3+^ ions and four-photon NIR emission [59].

## 3. Synthesis Strategy and Surface Modification of UCNPs

The luminescence properties of upconversion nanomaterials are closely related to their preparation methods. Different preparation methods will affect the size, morphology, and corresponding microstructure of luminescent materials, making their application directions more diversified. Among them, thermal decomposition, hydrothermal decomposition, and co-precipitation are the three most commonly used methods, and their advantages and limitations are shown in Table 1. Other synthesis methods, including sol-gel method and combustion method, are also discussed for comparison [60,61,62,63].

### 3.1. Thermal Decomposition Method

The thermal decomposition method is based on the high-temperature decomposition of organometallic precursors (such as metal trifluoroacetate) in high-boiling organic solvents (such as 1-octadecene). Surfactants are long-chain hydrocarbons and functional groups, such as -COOH, -NH_2_ or -PO_3_H (such as oleic acid, oleyl amine, trioctyl phosphine and trioctyl phosphine oxide), which are used as ligands, thus preventing the aggregation of nanoparticles. This method was originally developed by Professor Yan and others for the synthesis of LaF_3_ nanoparticles [64], and later extended to the synthesis of high-quality NaYF_4_ upconversion nanoparticles [65]. Capobianco et al. used trifluoroacetate precursors to synthesize Yb, Er and Yb, Tm co-doped NaYF_4_ nanoparticles [101,102]. The asymmetric molecular structure of octadecene (with a high boiling point of 315 °C) is used as a solvent, and oleic acid is used as a passivation ligand. Based on the separation of nucleation and growth of nanocrystals, an improved method has been developed [100], that is, the precursor is slowly added, followed by heating, to synthesize highly monodispersed nanoparticles. Using the same strategy, Murray and colleagues prepared hexagonal NaYF_4_:Yb,Er nanoparticle bodies, and precisely controlled their morphology and size [103]. Lim et al. The NaGdF_4_:Er^3+^/Yb^3+^ colloidal particles with an average particle size of 32 nm were successfully synthesized by the thermal decomposition method. The prepared phosphor shows bright green upconversion luminescence under a 976 nm semiconductor laser. Phosphor particles increase the scattering of optical coherence tomography (OCT) scanning radiation, thereby observing higher image contrast [104]. So far, this method has been widely used to synthesize a series of upconversion fluoride and oxyfluoride nanoparticles (Table 1). Although this method can produce high-quality upconversion nanoparticles, it also has some disadvantages, including expensive materials, air-sensitive precursors, and the production of toxic byproducts (such as hydrogen fluoride). Recently, the Pu team reported a safe and environmentally friendly method to replace octadecene (ODE) with paraffin liquid as a high-boiling non-coordinating solvent [105]. This method is biologically cheaper and sustainable.

### 3.2. Hydrothermal Method

Hydrothermal synthesis is another cheap method to obtain high-quality nanocrystals. This process is usually carried out at high pressure and high temperature above the boiling point or even the critical point. A special container called autoclave is used. The main disadvantages are the opacity of the reactor and the lack of in-situ reaction process and mechanism research. On the other hand, the advantages of high crystallinity, good dispersion, and no post-treatment make it one of the most popular synthesis technologies of lanthanum-doped nanoparticles. Organic ligands/surfactants such as oleic acid [106], ethylenediamine tetraacetic acid [107,108], cetyltrimethylammonium bromide [107], and polyethyleneimine [109] are added together with precursors to achieve synchronous control of size, morphology, crystalline phase, and surface properties. For example, Zhao and colleagues reported the hydrothermal process of oleic acid mediated upconversion NaYF_4_ nanocrystal synthesis, which has different shapes and morphology, including nanotubes, nanorods, and flower patterned nano disks [110]. Liu et al., used Ga^3+^ doping method to control the crystal phase [111]. They found that after on adding Ga^3+^ (accurately controlling the concentration), the required reaction temperature and time were greatly reduced, and the ultra-small NaYF_4_ upconversion nanoparticles underwent a rapid cubic to hexagonal phase transition. Li et al., demonstrated a synthesis strategy of multiphase, interface controlled monodisperse nanoparticles based on Liquid-Solid-Solid transfer and separation mechanism [112]. With some improvements, this method is also used for the preparation of upconversion nanoparticles with different fluorine and oxyfluorine contents (Table 1).

### 3.3. Co-Precipitation Method

Among the various methods of preparing nanocrystals, the co-precipitation method is the most promising technology, providing a convenient, safe, and economical method for preparing ultra-small and monodisperse upconversion nanoparticles, without the need for expensive equipment and toxic chemicals. These nanoparticles usually need post-treatment (calcination or annealing) to improve the crystallinity of the material. This method was first used by Veggel et al. for the synthesis of lanthanide ions (Eu, Er, Nd, and Ho) doped LaF_3_ downconversion nanoparticles [113]. Yi et al., subsequently demonstrated the application of this method in the preparation of upconversion nanoparticles. They used water-soluble precursors and octadecyl dithiophosphoric acid restriction ligands to prepare ultra-small (5 nm) monodisperse nanoparticles [89]. Guo et al. synthesized monodisperse NaYF_4_:Yb,Er upconversion nanoparticles of different sizes (37–166 nm) by adjusting the molar ratio of ethylenediaminetetraacetic acid to total lanthanides [114]. They also found that annealing these nanoparticles at a temperature of 400–600 °C can achieve a great increase in luminous intensity (up to 40 times). Recently, Huang et al.’s team prepared Sc^3+^-doped upconversion nanoparticles by lanthanide ion precipitation in the presence of oleic acid and octadecene [90]. They found that the crystalline phase transition depends on the volume ratio of oleic acid to octadecene, through the intermediate monoclinic/hexagonal coexisting phase (oleic acid:octadecene = 3:9), from the pure monoclinic phase Na_3_ScF_6_ (Oleic acid: octadecene = 3:17) to pure hexagonal phase NaScF_4_ (oleic acid: octadecene = 3:7). Due to the small radius of Sc^3+^, the emission of Na_x_ScF_3+x_:Yb,Er is quite different from that of NaYF_4_:Yb,Er, which can extend the application range of upconversion luminescent nanoparticles from optical communication to disease diagnosis.

### 3.4. Sol–Gel Method

The sol–gel method uses metal organics or inorganic salts as the matrix, generates gelatin through the process of hydrolysis and polycondensation, and then is dried or sintered to obtain UCNP. It has been successfully applied to the preparation of thin film coatings and glass materials for upconversion luminescent materials [115,116], but usually requires high-temperature post-processing to improve its crystallinity in order to obtain better luminescence effects. Prasad and colleagues first prepared erbium-doped ZrO_2_ nanoparticles using sol-latex-gel technology [115]. However, the size of UCNP is difficult to control, and the agglomeration after high temperature calcination brings some difficulties to the surface modification of the material and limits its biomedical applications [116,117,118,119,120,121].

### 3.5. Combustion Method

Different from the solvothermal method that uses lower temperature and longer reaction time, the combustion method provides a time-saving (a few minutes) method for the preparation of rare earth doped nanoparticles, and the reaction temperature is generally 500–3000 °C. This process takes place in the form of a combustion wave under a controlled explosion. The combustion method is based on a highly exothermic reaction. A variety of oxides and oxysulfide-containing single nucleotide chain nucleotides were synthesized by this method. For example, Zhang et al., used the glycine-nitrate process to synthesize monoclinic phase Gd_2_O_3_:Er^3+^ upconversion nanoparticles [122]. Dissolve the Gd_2_O_3_ and Er_2_O_3_ upconversion nanoparticles in diluted nitric acid, evaporate and heat. Rapid self-sufficient combustion will produce fluffy powder, which is heated to 600 °C for 1 h to remove nitrates and organic residues. The combustion method has the advantages of saving time and energy, but at the same time there is inevitably the phenomenon of aggregation of synthetic materials.

It is also worth mentioning that flame synthesis is another fast method for preparing upconversion nanoparticles. Ju et al., synthesized Y_2_O_3_:Yb,Er (or Tm,Ho) nanoparticles with a gas-phase flame one-step method, with an average size of less than 30 nm [123]. The results show that temperature has a strong influence on particle size, morphology, and photoluminescence intensity.

## 4. Surface Modification of UCNPs

The surface of UCNP obtained by various methods usually contains water-transporting organic ligands (amine oleate, octadecene, oleic acid, etc.). This makes UCNPs difficult to dissolve in water, which affects their biomedical applications to a certain extent. Therefore, it is necessary to modify the surface of UCNPs. So far, various surface modification methods have been reported, including silica coating, ligand exchange, ligand oxidation, ligand attraction, and layer-by-layer assembly [124,125,126,127,128,129,130,131,132,133], as shown in Figure 3.

### 4.1. Silica Coating

Surface silanization (or coated silica coating) is an inorganic surface treatment strategy that can make nanoparticles have water solubility and biocompatibility. It is known that silica is highly stable, biocompatible, and optically transparent. When used as a coating material, the method of surface silanization can flexibly provide abundant functional groups (such as -COOH, -NH_2_, -SH, etc.) to meet the various needs of binding with biomolecules. Wang’s group reported that the surface of UCNP is coated with polyhedral oligomeric silsesquioxane (POSS) to make the particles highly hydrophobic. They also described the preparation of liquid marbles based on optically and magnetically active dual-functional UCNPs, and their use as microreactors for the study of photodynamic therapy of cancer cells [134]. Veggel et al., reported the synthesis of SiO_2_-coated LaF_3_:Yb^3+^/Er^3+^ UCNPs using the Stöber method, and the thickness of the silica shell was controlled below 15 nm [135].

### 4.2. Ligand Exchange

Ligand exchange is a surface modification method that replaces the original hydrophobic ligands with some hydrophilic ligands without significantly affecting the chemical and optical properties of UCNP itself. Chow’s group prepared oil-soluble upconversion luminescent nanocrystals by pyrolysis, which are wrapped by oleylamine molecules. Then they exchanged ligands with dicarboxylic acid polyethylene glycol polymers and oleylamine molecules on the surface. Hydrophilic dicarboxylic acid PEG molecules can not only convert nanocrystals into water-soluble form, but also further couple the surface carboxyl groups with biomolecules [136]. Murray and colleagues reported the use of nitrosotetrafluoroborate (NOBF_4_) to replace the OA and OM ligands attached to the surface of nanoparticles, so that the nanoparticles can exist stably in a variety of polar media for a long time without producing aggregation or precipitation [137].

### 4.3. Ligand Oxidation

Ligand oxidation is another effective method to obtain water-soluble UCNP based on the selective oxidation of surface carbon-carbon double bonds (R−CH = CH−R’). This method requires the presence of unsaturated bonds in the original ligand. For example, the OA ligand on the surface of UCNPs can be oxidized to azelaic acid (HOOC(CH_2_)_7_ COOH), thereby making UCNP a hydrophilic material. Li’s group uses m-chloroperoxybenzoic acid as an epoxidizing reagent to oxidize the oleic acid ligand with carbon double bonds on the surface to a ternary epoxy compound, and then interact with organic molecules containing active functional groups (such as MPEG oh) to carry out a ring-opening reaction. Hydrophilic molecules are grafted onto oleic acid molecules to form the water-soluble conversion of luminescent nanoparticles [133]. Yan et al. used a clean and easily available strong oxidant ozone to oxidize the OA on the surface of UCNPs to azelaaldehyde or azelaic acid through ozone decomposition under certain conditions [138]. However, the ligand oxidation method has disadvantages such as long time and low efficiency, which is not conducive to actual production.

### 4.4. Ligand Attraction

The amphiphilic substance with both hydrophilic and hydrophobic functional groups acts on the surface of the nanoparticles. The hydrophobic functional groups in the substance are adsorbed by the oleic acid ligands on the surface of UCNPs through hydrophobic interaction, while the hydrophilic functional groups act as surface modification. Yi et al., first synthesized NaYF_4_:Yb/Er@NaYF_4_ upconversion luminescent nanoparticles with surface oleylamine modified by thermal decomposition method, using polyacrylic acid embedded with octylamine and isopropylamine as modifiers, using the hydrophobicity of polymer molecules. The hydrophobic interaction between the octyl and isopropyl groups of the UCNPs and the hydrophobic hydrocarbon groups on the surface of the UCNPs makes the polymer coat the surface of the UCNPs. The carboxyl groups in the polymer molecules make the hydrophobic UCNPs hydrophilic, realizing the resistance to UCNPs.

### 4.5. Layer-by-Layer Assembly

Layer-by-layer assembly involves the electrostatic adsorption of oppositely charged anions or cations on the UCNP surface. Electrostatic attraction is one of the strongest and most stable interactions known in nature. Specific layer-by-layer modification usually contributes to the specific biological applications of UCNP. The advantage of this method is that it can prepare coated colloids of different shapes and sizes, with uniform layers of different compositions and controllable thickness [139]. Most importantly, this method can control the surface potential, size, and incorporated functional group ligands of UCNP, which is very important for cell internalization and biological targeting. Polyacrylic acid (PAA) coated with Yi groups on the surface of upconversion luminescent nanocrystals by the layer assembly method. PAA contains 25% octylamino groups and 40% isopropylamine groups, which can transfer oil-soluble upconversion luminescent nanocrystals to the water phase and further couple with biomolecules [140].

In addition to the improvement of biocompatibility, surface modification can also improve the optical properties of UCNPs, laying a foundation for the further application of UCNPs in bioimaging and other fields.

## 5. Biological Applications of UCNPs

Compared with traditional luminescent materials (including organic dyes and quantum dots), UCNPs have the advantages of high chemical stability, good optical stability, and narrow band gap emission. In addition, under the excitation of near-infrared light, it has strong biological tissue penetration, no damage to biological tissue, high signal-to-noise ratio, and has been widely used in the biological field. This paper also focuses on the application of up conversion luminescent materials in biological imaging, biological detection, and photodynamic therapy [141,142].

For highly infiltrative cancers, including glioblastoma multiform (GBM), it cannot be resected completely. Routinely, the conventional direct introduction to UCNPs suspension into the tissue will cause a series of biocompatibility problems. Thus, implantation of optical fiber has a heavy burden on the patient’s body and a high risk of infection. Daniel et al. [143] reported the fabrication of an optical-guided UCNPs implant. Comparatively, the highly biocompatible UCNPs implant is placed into the brain via craniotomy and sealed. Then, the NIR light source passes through the healed scalp and points in the implant to stimulate the UCNPs implant, which will emit visible light to target the photosensitive metabolite protoporphyrin-IX (PpIX) in brain tumors. The flexible light guide with FEP coating approved by the FDA can also maintain NIR to visible light spectrum transduction when the implant is bent to 90°, as shown in Figure 4a. The tumor size of PDT-treated mice was significantly smaller than untreated mice, as shown in Figure 4b. Implant-based UCNPs also allow them to recover from the tissue when they are no longer needed, which cannot be achieved by direct injection of UCNPs suspension into the tissue. A wide range of emission spectrum can be engineered into UCNPs so that the implant can activate multiple drugs simultaneously without cross-interference.

Wang et al. [144] reported a method that can change intracellular pH UCNP@ZIF + TPP + PA nanometer material. The intracellular pH is usually weakly alkaline, which is not conducive to the application of acid-responsible nanomaterials. The proposed new nanomaterials (UCNP@ZIF + TPP + PA) were designed with NaYF_4_: Yb: Tm (UCNPs) inner core coated by the framework of porous imidazolium zeolite (ZIF-8) for co-loading of photoacid (PA) and acid-responsive porphyrin (TPP). With 980 nm laser irradiation, the emission light of UCNPs activated PA to release H^+^. In the new acidic environment, TPP was protonated to increase its water solubility and reduce its aggregation. Simultaneously, transformed UV Vis also activates protonated TPP, which produces more singlet oxygen (^1^O_2_) to kill cancer cells, to enhance the therapeutic effect of photodynamic therapy (PDT); the synthesis route and therapeutic principle are shown in Figure 5.

One of the key challenges in the process of PDT therapy is the accurate killing of cancer cells without destroying normal cells to achieve the desired therapeutic effect. Li et al. [145] designed a type of upconversion nanoprobes (mUCNPs) for intracellular cathepsin B (CAB) reactive PDT, composed of multi-shell upconversion nanoparticlesNaYF_4_: Gd@NaYF_4_:Er, Yb@NaYF_4_: Nd, Yb), with the function of in situ self-tuning therapy effect prediction, as shown in Figure 6.

Similarly, the Zhang group [146] designed an amplifier with multiple upconversion luminescence, composed of photo-caged DNA nano-combs and upconversion nanoparticles (UCNPs) sensitized with IRDye^®^ 800CW, to realize the near-infrared light switch cascade reaction triggered by specific microRNA and accurate photodynamic therapy for early cancer. Under 808 nm light irradiation, the generated ultraviolet light cuts off the “photo-zipper” to induce the cascade hybridization reaction of the microRNA response. This activates the photosensitizer connected to different hairpins to produce reactive oxygen species (ROS) under the blue light emitted simultaneously, to carry out effective PDT, as shown in Figure 7. The amplifier showed desirable serum stability, excellent controllability of reactive oxygen species generation, high specificity for target cancer, and sensitivity to specific microRNA expression. In vivo and in vitro experiments showed strong inhibition on cell proliferation, strong ability to induce apoptosis of tumor cells, and distinct inhibition of tumor growth.

Lin et al. [147] designed a spindle-like UCNPs nanoprobe, coated with a layer of gold nanoparticles to enhance upconversion luminescence (UCL), as shown in Figure 8. The results of biocompatibility, blood routine, bioimaging, and anti-cancer tests showed that it was easier for the spindle-like nanoprobes to enter biological tissues. In addition, the combination of SPS@Au and ZnPc (SPSZ) is a potential candidate for synergistic immune photodynamic therapy (PDT), with enhanced UCL effect and excellent biocompatibility.

The Li group [148] designed core-satellite metal-organic through electrostatic self-assembly framework@UCNP superstructures, composed of a single metal-organic framework (MOF) NP as the core, and Nd^3+^-sensitized UCNPs as the satellites. In vitro and in vivo experiments show that the double photosensitizer superstructure has a three-mode (magnetic resonance/UCL/ fluorescence) imaging function and excellent anti-tumor effect under the excitation of 808 nm near-infrared light, avoiding overheating caused by laser irradiation. After being exposed to an 808 nm laser for 5 min, the temperature of the irradiated area was lower than 42 °C, and without damage to mice. However, under the same conditions, a 980 nm laser can heat the irradiated area to above 50 °C and severely burn the skin. These findings indicate that 808 nm excitation has a much weaker tissue thermal effect and is more suitable for biological applications.

Sun et al. [149] designed and prepared one kind of lanthanide (Ln^3+^)-doped upconversion nanocomposites with multi-functions, which can not only provide temperature feedback in PTT process, but also play the photodynamic therapy (PDT) function for the synergistic effect of tumor therapy. Based on NaYF_4_:Yb, Er upconversion nanoparticles (UCNPs), mesoporous SiO_2_ was modified on the surface combined with photosensitizer Chlorin e6 (Ce6) molecules, which could be excited by red emission of Er^3+^ under the 980 nm laser. Cit-CuS NPs were further linked on the surface of the composite as a photothermal conversion agent, therefore, the temperature of the PTT site can be monitored by recording the ratio of I_525_/I_545_ of green emissions, especially within the physiological range, as shown in Figure 9. Based on the guidance obtained from spectral experiments, they further investigated the dual-modal therapy effect both in vitro and in vivo, respectively, and acquired decent results.

In addition, these rare earth doped nanoparticles also have strong scintillation luminescence that can be used for X-ray induced photodynamic therapy, which is a very hot area, as this new therapy can be used for deep as well as skin cancer treatment [150,151,152,153,154,155,156,157,158,159,160,161,162,163].

## 6. Summary and Perspective

Over the last decades, UCNPs have made remarkable advances in the treatment of critical diseases, greatly promoting the application of modern precision medicine in the life system with its enhanced therapeutic effect, high space-time controllability, deep tissue penetration, and minimal invasion. However, despite the remarkable achievements, there are still some challenges of UCNPs. (1) The stability of luminous efficiency after surface modification: in the surface modification, the oil-soluble molecules will be modified to improve biocompatibility. Nevertheless, the dispersing ability of UCNPs in oil and water is different. After surface modification, it is easy to cause fluorescence quenching and reduce the upconversion efficiency. (2) The biological toxicity of UCNPs: many studies have shown that the reasonable optimization of chemical composition, particle size distribution, and surface modification can significantly improve the biocompatibility of UCNPs, which can be used in biomedical applications. However, there are no tests to evaluate its long-term toxicity, including potential immune response and mutagenic effect. (3) Technical gaps in clinical trials: so far, UCNPs-based phototherapy has not been applied to human beings, mainly due to biosafety or therapeutic effect. There is still a long way to go from laboratory animals to human-level technical standard updates. In summary, UCNPs offer a tremendous opportunity to practice precision medicine. We expect that stable surface modification, low toxicity, and clinical trials will make UCNPs more competitive in the biological field.

## Figures and Tables

**Figure 1 nanomaterials-11-02474-f001:**
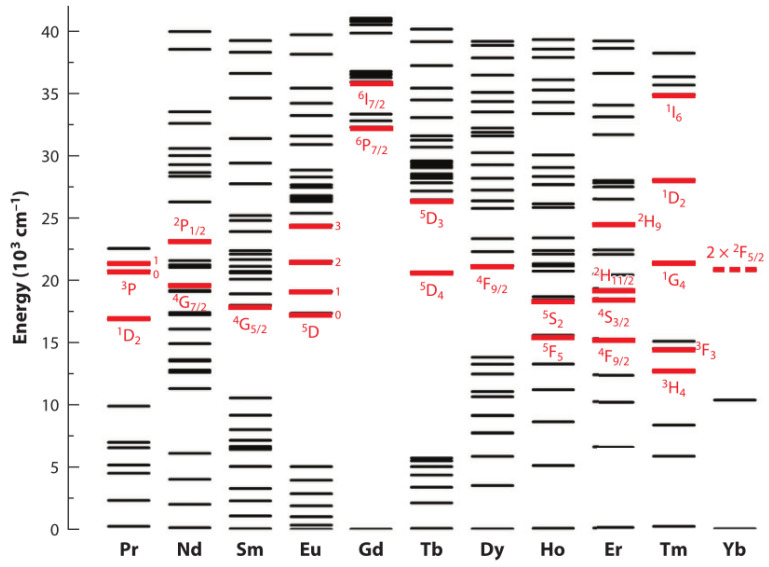
Energy-level diagrams of rare-earth ions. Typical upconversion emissive excited states are highlighted by a red bold line. Reprinted with permission from ref. [11]. Copyright 2015 Annual Reviews.

**Figure 2 nanomaterials-11-02474-f002:**
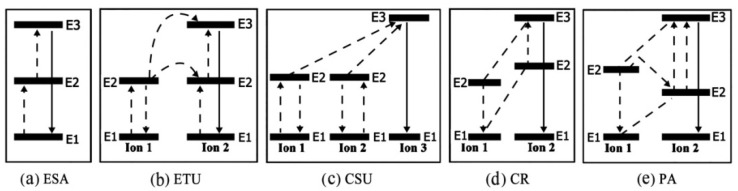
Upconversion mechanisms of RE-doped UCNPs: (**a**) ESA, (**b**) ETU, (**c**) CSU, (**d**) CR and (**e**) PA.

**Figure 3 nanomaterials-11-02474-f003:**
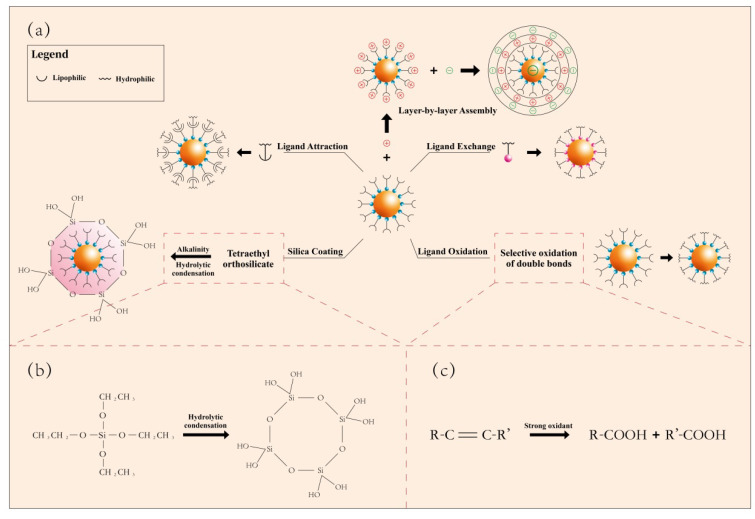
(**a**) Schematic flow diagram of different surface modification methods (silica coating, ligand exchange, ligand oxidation, ligand attraction, and layer–by–layer assembly). (**b**) Chemical reaction formula for preparing silica coating with tetraethyl orthosilicate. (**c**) Basic reaction formula of ligand oxidation process.

**Figure 4 nanomaterials-11-02474-f004:**
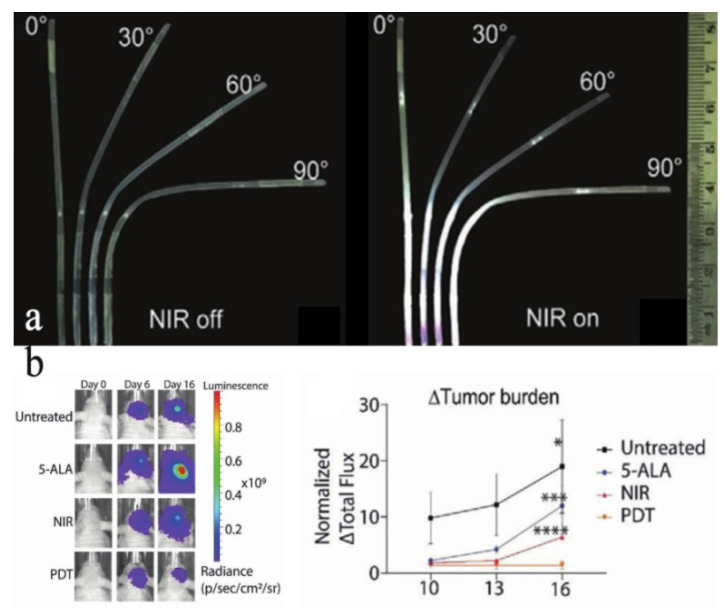
(**a**) UCNPs implant when not excited with NIR and emission intensities visualization at different angles of UCNPs implant bending, excited with 1583 mW cm^−2^ of NIR. (**b**) IVIS imaging indicates that PDT mouse tumors were regressing, as compared to other control groups. The normalized change of tumor burden in all experiment groups over time (*n* = 5 mice/group. * *p* = 0.0327, *** *p* = 0.0002, **** *p* < 0.0001. Two-way analysis of variance (ANOVA) with Bonferroni’s multiple comparison test), reprinted with permission from ref. [143]. Copyright 2020 John Wiley and Sons.

**Figure 5 nanomaterials-11-02474-f005:**
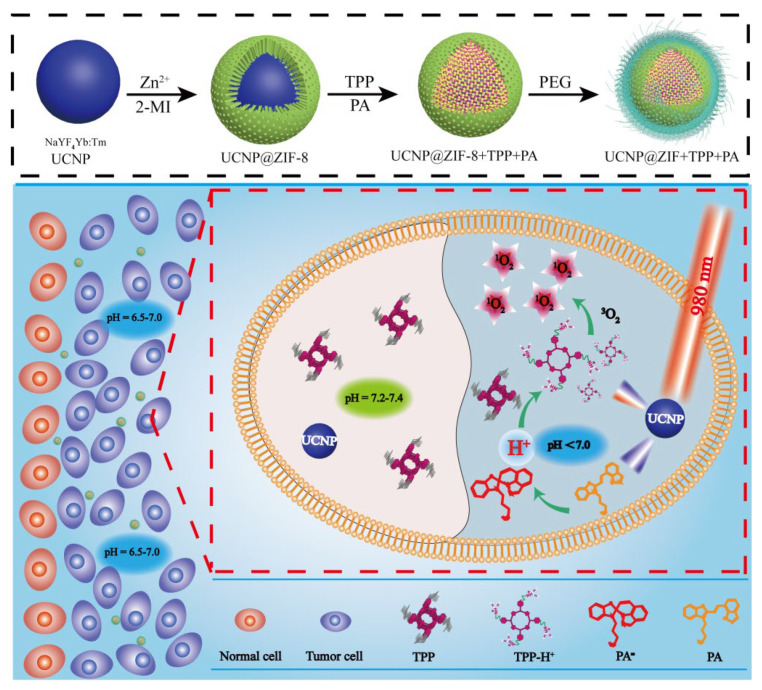
Synthetic route and anticancer mechanism of UCNP@ZIF + TPP + PA. Responding to weak acid tumor microenvironment, UCNP@ZIF + TPP + PA released TPP. When acid-responsive TPP entered the weak alkaline cell, it aggregated again. With 980 nm light irradiation, UCNP emitted UV-Vis light. Photoacid absorbed UV-Vis light and changed its structure to produce H+, which restructured the intracellular pH value. In the new weak pH, the aggregation of TPP was decreased by its protonation. Meanwhile, protonated TPP was activated by the transformed UV-Vis light and produced more ^1^O_2_ for enhancing PDT, reprinted with permission from ref. [144]. Copyright 2014 Royal Society of Chemistry.

**Figure 6 nanomaterials-11-02474-f006:**
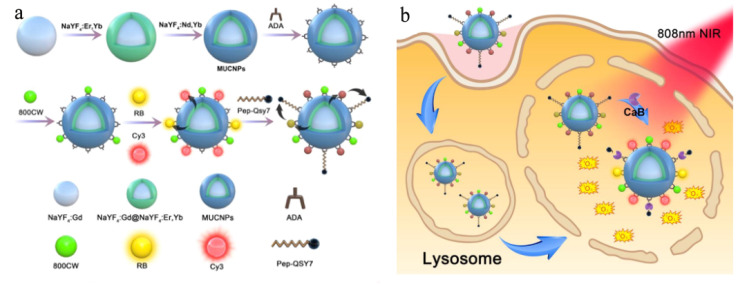
Schematic illustrations of (**a**) synthesis of the upconversion nanoprobe and (**b**) Intracellular CaB-Activated PDT with CaB imaging for Therapeutic Effect Prediction, reprinted with permission from ref. [145]. Copyright 2020 American Chemical Society.

**Figure 7 nanomaterials-11-02474-f007:**
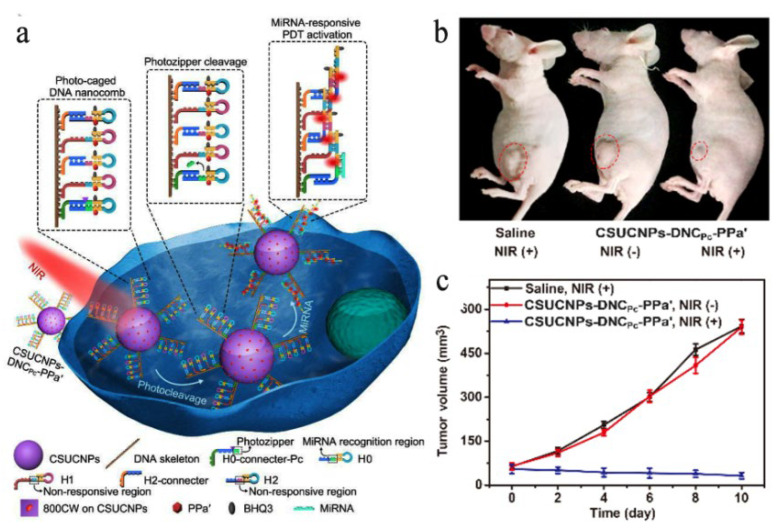
(**a**) Schematic illustration of NIR photo-switched miRNA amplifier for precise PDT. (**b**) Representative images at day 10 and (**c**) tumor volumes of early-stage breast cancer-bearing mice, treated with saline, CSUCNPs-DNC’Pc-PPa’ and CSUCNPs-DNCPc-PPa’ before and after 808 nm light irradiation at 1 W/cm^2^. Error bars indicate means ± SD (*n* = 5), reprinted with permission from ref. [146]. Copyright 2020 John Wiley and Sons.

**Figure 8 nanomaterials-11-02474-f008:**
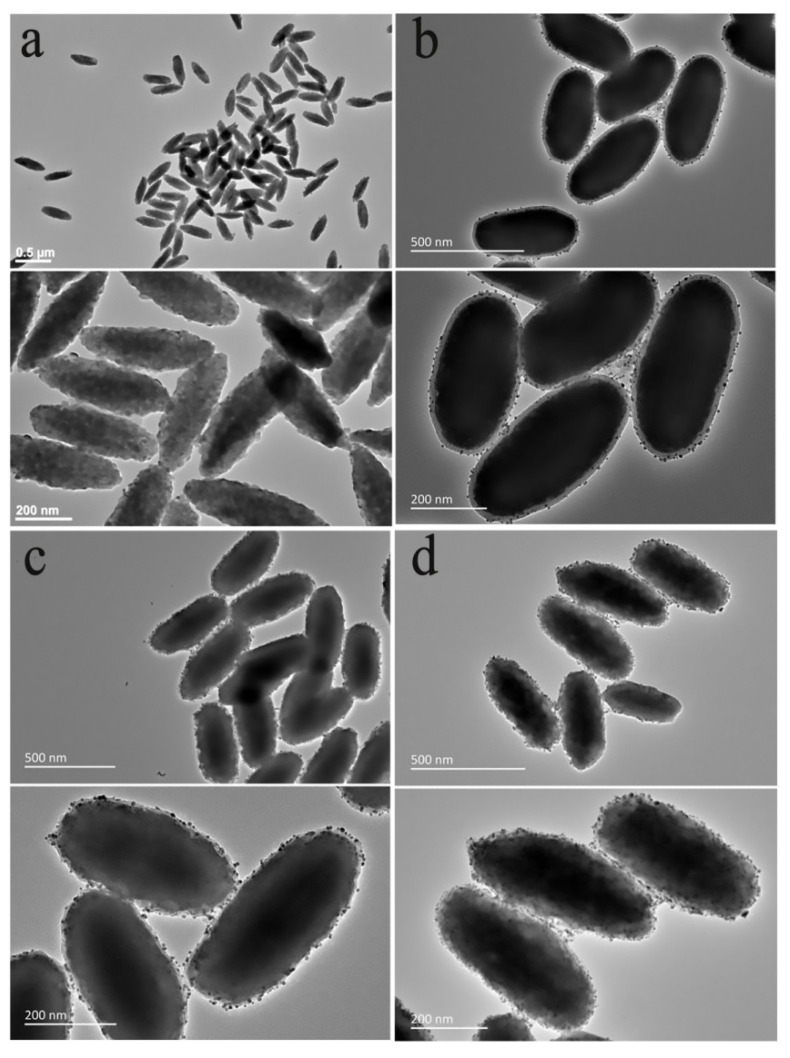
TEM image of (**a**) spindle precursor and (**b**) SPS@Au (LBL1) and (**c**) SPS@Au (LBL2) and (**d**) SPS@Au (LBL3). Reprinted with permission from ref. [147]. Copyright 2020 American Chemistry Society.

**Figure 9 nanomaterials-11-02474-f009:**
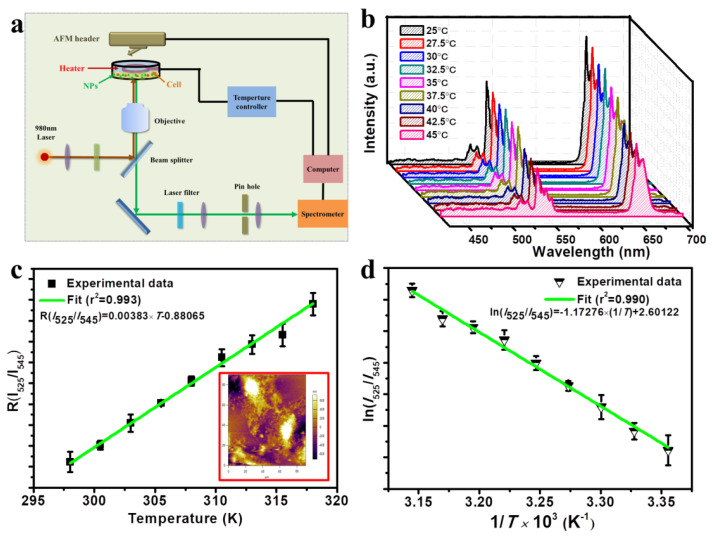
(**a**) Schematic diagram of the detection of the temperature and emission spectrum of UCNPs-Ce6@mSiO_2_-CuS incubated with cells in physiological range; (**b**) UCL emission spectrum of UCNPs-Ce6@mSiO_2_-CuS incubated with cells at different temperatures by external heating. The peaks were normalized at 525 nm; (**c**) FIR of the green UC emissions for the ^2^H_11/2_/^4^S_3/2_ → ^4^I_15/2_ transitions relative to the temperature of UCNPs-Ce6@mSiO_2_-CuS incubated with cells. The inset picture is the AFM image of the cell after spectral detection; (**d**) A plot of ln(I_52_5/I_545_) versus 1/T to calibrate the thermometric scale for UCNPs-Ce6@mSiO_2_-CuS incubated with cells. Reprinted with permission from ref. [149]. Copyright 2019 Elsevier.

**Table 1 nanomaterials-11-02474-t001:** Advantages and disadvantages of upconversion nanoparticles synthesized by different methods and examples.

Method	Advantages	Disadvantages	Examples
Thermal Decomposition	Large product volume; small size distribution	The equipment is expensive; the precursor is sensitive to air; toxic by-products	ReF_3_ (Re = Y,La) [64] NaLuF_4_ [65] NaYbF_4_ [65] MF_2_ (M = Ca,Sr,Ba) [66] LiYF_4_ [67] NaGdF_4_ [68] BaREF_5_ (RE = Y,Gd) [69] KY_3_F_10_ [70] RE_2_O_3_ (RE = Y,La,Gd) [71] REOF (RE = Y,La,Gd) [72]
Hydrothermal Decomposition	Inexpensive precursors; no need for post-processing; precise size and shape control	Need an autoclave; the reaction process is unobservable and uncontrollable	REF_3_ (RE = Y,La,Ce,Gd) [73] MF_2_ (M = Ca,Sr,Ba) [74] NaYF_4_ [75] NaLaF_4_ [76] NaLuF_4_ [77] KMnF_4_ [78] BaGdF_5_ [79] RE_2_O_3_ (RE = Y,Gd,Er) [80] LaOF [81] REPO_4_ (RE = Ga,Yb,Lu) [82]
Co-precipitation	Fast synthesis speed; inexpensive equipment and safe precursors	Need post-processing	NaYF_4_ [83] NaGdF_4_ [84] NaTbF_4_ [85] NaLuF_4_ [86] KGdF_4_ [87] CaF_2_ [88] LaF_3_ [89] NaScF_4_ [90]
Sol-gel Method	Inexpensive precursors; small product size	The precursor preparation process is complicated; product is easy to reunite	BaIn_2_O_4_:Yb^3+^/Tm^3+^/RE^3+^ (RE = Er^3+^, Ho^3+^) [91] NaPbLa (MoO_4_)_3_: Er^3+^/Yb^3+^ [92] La_4_Ti_3_O_12_ [93] Gd_2_O_3_: Er^3+^/Yb^3+^/Bi^3+^ [94] CaTi_4_O_9_: Er^3+^/Yb^3+^ [95]
Combustion Method	Fast synthesis speed; energy saving; controllable product quantity	Expensive equipment; high temperature; the particle size of the material is large and easy to agglomerate	Ba_5_ (PO_4_)_3_OH: Er^3+^/Yb^3+^ [96] Na_3_Y (PO_4_)_2_: Er^3+^/Yb^3+^ [97] BaLaAlO_4_:Er^3+^/Yb^3+^ [98] ZrO_2_: Ho^3+^/Yb^3+^ [99] LaO_3_: Er^3+^/Tm^3+^ [100]

## Data Availability

The study did not report any data.

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
