# Peer review of "Recent Advances of Upconversion Nanomaterials in the Biological Field"

_nanomaterials, 2021, doi:10.3390/nano11102474_

Round 1

Reviewer 1 Report

The authors made an effort doubling the number of references adding adequate relevant ones.

Based on this consideration, I am ready to accept their paper.

Best regards.

Reviewer 2 Report

The authors have done a great and extensive job to improve the quality of the manuscript.

They have dealt with all my comments and introduced enough detail in the revised version.

This manuscript is a resubmission of an earlier submission. The following is a list of the peer review reports and author responses from that submission.

Round 1

Reviewer 1 Report

Dear Editor,

Please, find following my comments concerning Paper nanomaterials--1320472- by Cunjin Gaoet al., untitled “  Recent Advances of Upconversion Nanomaterials in Biological  Field

This review paper is interesting but does not compare with the already published ones this year in Coordination Chemistry Reviews (2021), 440, 213971; Coordination Chemistry Reviews (2021), 429, 213642Materials Chemistry Frontiers (2021), 5(4), 1743-1770, Analyst (Cambridge, United Kingdom) (2021), 146(1), 13-32, or in 2020 Journal of Luminescence (2020), 228, 117627

Best regards

Reviewer 2 Report

I find of very little interest for the readers this review paper in its present form. In addition, sections 2 and 3 are unclear; the readers can extract very little information from them.

Major comments:

- Section 2 must be rewritten in a clearer and more attractive way. For example, it must be clarified the requirements for ion 1 and ion 2 (or ion 3). In addition, at least one example must be provided for each mechanism.

- Section 3 is written in a very general way. Also, it is unclear. For example, it is written that “the co-precipitation method is the most promising technique..”. However, it seems that just shortcommings are presented related to this approach.

- Section 3.2. “Surface Modification of UCNPs” must be more systematically presented and a diagram including the different procedures must be depicted.

- Another major comment is that the review is entitled “…. Upconversion Nanomaterials…” However, there is not discussion about sizes and their effect.

Minor comments:

  • Second paragraph of intro: “and NO absence of self-fluorescence interference,” means “presence of self-fluorescence interference?”
  • The term “RE” is used in the second paragraph of the intro. However, it is not defined up to Section 3.
  • Section 3.1.2. It must be provided a reference for the “Hydrothermal Method”
  • The first paragraph under section 4, is very similar to the first paragraph under section 4.1. Please unify and delete repetitions.
  • 1.3. Co-Precipitation Method. I do not understand the word “Conversely” within the context